# Alfalfa leaf weevil larvae and adults feeding induces physiological change in defensive enzymes of alfalfa

Hui Liu [1,2], Xuzhe Wang[1], Yong Ma[1], Wanshun Gao[1], Chunhui Ma[1] *

1 College of Animal Science and Technology, Shihezi University, Shihezi, Xinjiang, China, 2 College of Life Science, Shihezi University, Shihezi, Xinjiang, China

* chuihuima@126.com

**Data Availability Statement:** All relevant data are within the paper and its Supporting information.

**Funding:** This research was funded by the National Modern Agriculture Industry Technology System (CARS-34) of the Ministry of Agriculture and Rural

## Abstract

When insects harm plants, they activate relevant enzyme systems for defense, and changes in enzyme activity, to a certain extent, reflect the host plant's ability to resist insect damage. Alfalfa leaf weevils (*Hypera postica* Gyllenhal) are the main economic insect pest of alfalfa, which seriously affect its yield and quality. To clarify the effects of feeding induction by alfalfa leaf weevil larvae and adults on defense enzymes in alfalfa, 'Zhongmu No. 1' variety was used as the experimental material. Comprehensive correlation analysis and principal component analysis were used to evaluate the corresponding patterns of 12 physiological indicators of alfalfa induced by insect feeding of different densities. Results showed that after feeding induction by adult and larval alfalfa leaf weevils, total antioxidant capacity (T-AOC), malondialdehyde (MDA), phenylalanine ammonia-lyase (PAL), tyrosine ammonia lyase (TAL), lipoxygenase (LOX), chymotrypsin inhibitors (CI), trypsin inhibitor (TI), and jasmonic acid (JA) in the alfalfa leaves increased with increasing feeding time. However, activities of catalase (CAT), peroxidase (POD), superoxide dismutase (SOD) and polyphenolic oxidase (PPO) in alfalfa leaves first increased and then decreased, showing a downward trend.

## Introduction

Alfalfa (*Medicago sativa* L.), a perennial herbaceous plant belonging to the legume family, is an important high-quality forage for herbivores, such as cows, sheep, pigs and special animals (rabbits, ostriches, deer, etc). It has a cultivation history of more than two thousand years in China. Alfalfa has a high nutritional value, is rich in proteins, cellular solutes, minerals, vitamins, and is known as the "King of Grass" [1]. As a high-quality forage with abundant nutrients, it is favored by herbivorous insect pests. There are at least 1000 species of arthropods in American alfalfa fields, of which 100~150 can cause harm [2]. Alfalfa leaf weevil (*Hypera postica* Gyllenhal), belonging to the order Coleoptera and Curculionidae family, is the main economic pest of alfalfa [3,4]. Both larvae and adults of alfalfa leaf weevils damage alfalfa crop. The generation overlap of alfalfa leaf weevils leads to growth inhibition, delayed maturity, decreased yield, and reduced competitiveness with weeds. In severe cases, it can cause alfalfa

Affairs. The funders had no role in study design, data collection and analysis, decision to publish, or preparation of the manuscript.

**Competing interests:** The authors have declared that no competing interests exist.

plants to wither, not bloom, and even die [5,6]. In addition, alfalfa leaf weevils can cause a decrease in the digestibility of crude protein and dry matter, affecting its quality [7,8].

To cope with the threat of herbivorous insects, plants have developed complex and diverse defense systems over a long period of evolution [9]. The defense of plants involves complex physiological and biochemical processes, including perception of herbivorous insects, triggering of early signaling events, activation of hormone signaling pathways, and synthesis of defense compounds [10]. When subjected to pest stress, plants activate relevant enzyme systems for defense, including peroxidase (POD), superoxide dismutase (SOD), catalase (CAT), polyphenolic oxidase (PPO), and phenylalanine ammonia-lyase (PAL) [11]. To some extent, the changes in enzyme activity of this type reflect the host plant's ability to resist insect pests [12,13]. Insect feeding can damage plant tissues, leading to the oxidation of chlorogenic acid in leaves by PPO and POD to produce highly active quinones [14,15]. These quinones can also undergo alkylation reactions with substances containing nucleophilic groups (such as—SH and—$NH_2$), inhibiting insect digestion of plant proteins and, to some extent, resisting insect harm [16,17]. Additionally, insect damage to plant tissues can trigger the production of reactive oxygen species (ROS). The accumulation of ROS can damage cells, and the plant's reactive oxygen species scavenging system is mainly composed of PPO, POD, SOD, and CAT. It can maintain the dynamic balance of ROS by clearing excessive $H_2O_2$ and superoxide anions, thereby protecting plants from harm [18–20].

Phenylalanine ammonia-lyase (PAL) is a rate-limiting enzyme in the phenylpropanoid metabolism pathway. When plants are affected by pests, the mechanism initiates or strengthens phenylpropanoid metabolism, increasing PAL activity in the affected area and leading to a large accumulation of lignin in the cell wall and thickening of the cell wall, thereby preventing the spread of pests. At the same time, the enhancement of PAL activity can also increase the phytochemical content, thereby avoiding and poisoning herbivorous insects and reducing pest damage. Studies have shown that herbivorous insects can increase PAL activity through feeding [21]. Thrips (*Frankliniella occidentalis*) can alter the expression levels of plant protective enzyme genes and the activity of enzymes in its own body, thereby improving adaptation to leguminosae plants [22]. Protease inhibitors (PIs) are a class of small molecular weight proteins widely present in plants that can inhibit the activity of proteases in insects and have a significant inhibitory effect on the growth and development of larvae [23]. PIs can also bind to proteases in the digestive tract of insects to form enzyme inhibitor complexes, blocking or weakening the protein hydrolysis of digestive enzymes, thereby affecting the digestion, absorption, and utilization of food by insects [24]. Chymotrypsin inhibitors (CIs) are widely present in leguminous plants, and their main function is to stop, block, or reduce the activity of chymotrypsin, thereby reducing protein degradation in the intestine, affecting the generation of amino acids and peptides, thus affecting the normal metabolism of organisms. Tyrosine ammonia lyase (TAL), lipoxygenase (LOX), and the plant hormone jasmonic acid (JA) play a core role in plant defense against herbivorous insects [25].

This study aimed to determine the effects of feeding induction by adult and larval alfalfa leaf weevils on the defensive substances of alfalfa to clarify the dynamic interaction between alfalfa leaf weevil feeding and alfalfa defensive properties.

## Materials and methods

### Materials

Alfalfa ('Zhongmu No. 1') was sown on April 19, 2022, in a flowerpot (220 mm × 190 mm) containing a mixture of field soil, floral soil, and perlite (2:1:1). On April 25, 2022, 10 alfalfa plants were kept in each flowerpot and placed in insect cages (120 nylon mesh, 75 cm × 75

cm × 75 cm). Watering alfalfa every 2–3 days and 100 mL/pot of Hoagland nutrient solution every 7 days to ensure normal growth of seedlings. After 6 weeks, healthy alfalfa with consistent growth was selected for insect infestation treatment.

Adult alfalfa leaf weevils were collected from fields in Shihezi, China. They were raised in insect cages (120 nylon mesh, 75 cm × 75 cm × 75 cm) with potted alfalfa and continuously bred for multiple generations. The 3rd-instar larvae and adults of the alfalfa leaf weevil with consistent vitality were selected and starved for 24 h before testing.

## Method

The experimental design for alfalfa feeding by alfalfa leaf weevil larvae and adults is shown in Fig 1. Each treatment had five flowerpots (replicates), and each flowerpot was placed in an insect cage (120 nylon mesh, 75 cm × 75 cm × 75 cm). All treatments were performed in the same experimental environment in outdoor at June 7–9, 2022. After being induced by alfalfa leaf weevil feeding for 24, 36, 48, 60, and 72 h (Fig 1), the injured alfalfa leaves were randomly selected from the top, middle, and bottom of the alfalfa plants. The injured alfalfa leaves were wrapped in tin foil, and frozen in liquid nitrogen for enzyme activity and other indicator

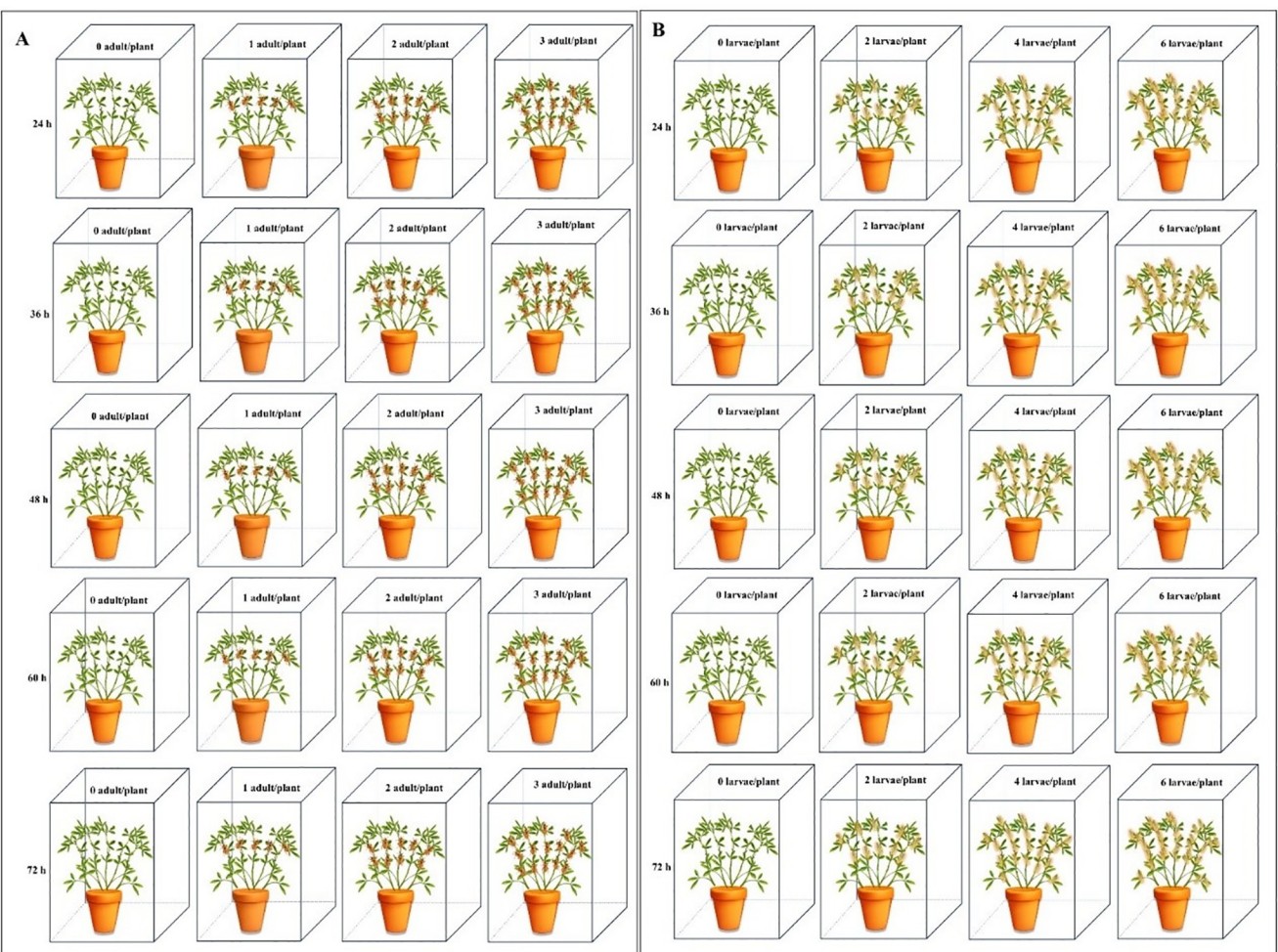

**Fig 1. Design of experiments.** A. Adult feeding, B. Larvae feeding, each treatment had five flowerpots (replicates).

measurements. Injured alfalfa leaves were grinded into powder using liquid nitrogen and 0.1 g of alfalfa leaf sample was taken (Analytical balance XB220A, Precisa Gravimetrics AG, Dietikon, Switzerland), added 1 mL of phosphate buffered saline solution (pH 7.4), and mixed well in an ice bath, and then centrifuge at 8000 rpm at 4˚C for 10 minutes (High-speed frozen centrifuge TGL16M, Shanghai Lu Xiangyi Centrifuge Instrument Co., Ltd., Shanghai, China). After that, supernatant was collected and placed in an ice box for testing. All enzyme activity measurements were performed in accordance with the kits manufacturer's instructions, which purchased from Shanghai Enzyme-linked Biotechnology Co., Ltd., Shanghai, China (Supporting information).

## Data statistics and processing

Data were preliminarily organized using Microsoft Excel 2010 software (Microsoft Corporation, Redmond, Washington, USA), and statistical analysis was conducted using SPSS 26.0 software (SPSS Inc., IBM Company, Chicago, IL, USA). The significance of the differences between treatments was analyzed using Duncan's multiple comparison test ($P < 0.05$). Perform principal component analysis using Pearson correlation coefficient. The effects of different insect densities and feeding times on various alfalfa leaf indicators were analyzed using Two-way ANOVA and plotted using OriginPro 2021 software (OriginLab, Northampton, MA, USA).

## Results

### Effect of defensive substances in alfalfa leaves induced by adult alfalfa leaf weevils

**Antioxidant enzymes activity in alfalfa leaves.** The effect of feeding induction by adult alfalfa leaf weevils on antioxidant enzymes in alfalfa leaves is shown in Fig 2. As shown in Fig 2A, after feeding induction by adult alfalfa leaf weevils, the CAT activity in alfalfa leaves was lower than that in healthy alfalfa. The difference in CAT activity of alfalfa leaves after feeding induction by adult alfalfa leaf weevils at 36, 48, 60, and 72 h was significant ($P < 0.05$). As the feeding time increased, the CAT activity of alfalfa leaves in each treatment showed a trend of first decreasing and then increasing. CAT activity at one pest/plant and two pest/plant was the lowest after 60 h of feeding induction, which was 34.58% and 28.94% lower than that of healthy alfalfa leaves, respectively. The CAT activity at three pests/plant was the lowest after 48 h of feeding induction, which was 26.78% lower than that of healthy alfalfa leaves. As shown in Fig 2B, the POD activity in healthy alfalfa leaves was significantly higher than that induced by feeding on adult alfalfa leaf weevils ($P < 0.05$). The POD activity of alfalfa leaves induced by feeding on adult alfalfa leaf weevils showed a trend of first decreasing and then increasing. After 60 h of feeding induction, POD activity was the lowest for one pest/plant and two pest/plant, which were 28.65% and 30.73% lower than that of the healthy alfalfa, respectively. After 48 h of feeding induction, POD activity was the lowest in the three pest/plant treatments, which was 23.78% lower than that of healthy alfalfa.

After feeding induction for 24 and 36 h, the SOD activity in alfalfa leaves was higher than that in healthy alfalfa. The SOD activity after feeding induction for 48, 60, and 72 h was significantly lower than that in healthy alfalfa ($P < 0.05$) (Fig 2C). The SOD activity of the one pest/plant and two pest/plant treatments was the lowest at 60 h, which was 19.71% and 18.53% lower than that of healthy alfalfa, respectively. The three pest/plant treatment reached its lowest point at 48 h of feeding, which was 13.99% lower than that of the healthy alfalfa. As shown in Fig 2D, after feeding induction by adult alfalfa leaf weevils, the T-AOC of alfalfa leaves was

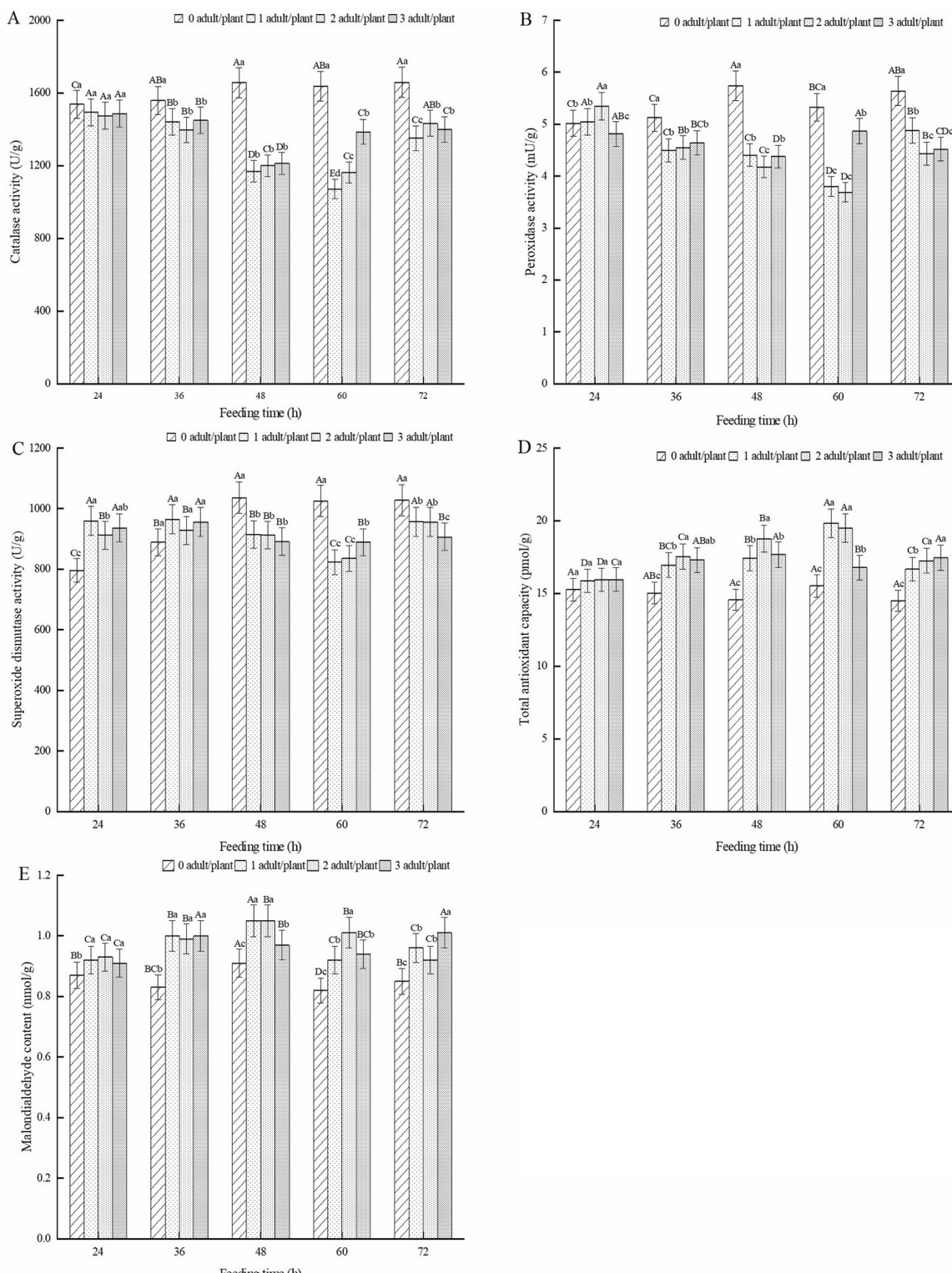

**Fig 2. Effect of antioxidant enzymes in alfalfa leaves induced by adult alfalfa leaf weevils.** Note: Different uppercase letters indicate significant differences in the same population density at different feeding times, whereas different lowercase letters indicate significant differences in population density at the same feeding time (P < 0.05, Duncan multiple comparisons).

higher than that of healthy alfalfa leaves, and the differences were significant at 36, 48, 60, and 72 h ($P < 0.05$). The T-AOC of alfalfa leaves induced by feeding with one and two pests/plant showed a trend of first increasing and then decreasing, and both reached their peak at 60 h of feeding, which was 27.84% and 25.61% higher than that of healthy alfalfa, respectively. The three pest/plant treatments showed a trend of first increasing and then decreasing, with the highest T-AOC at 48 h of feeding, which increased by 28.88% as compared with healthy alfalfa. As shown in Fig 2E, after feeding induction by adult alfalfa leaf weevils, the MDA content in alfalfa leaves was significantly higher than that in healthy alfalfa ($P < 0.05$). As the feeding induction time increased, the MDA content showed a trend of first increasing and then decreasing at 24, 36, 48, and 60 h of feeding. After 48 h of feeding induction, the MDA content of one pest/plant and two pest/plant reached the highest value, increasing by 15.52% and 15.69%, respectively, compared to healthy alfalfa. The three pest/plant treatments reached the highest value after 36 h of feeding, which was 21.05% higher than that of the healthy plants.

**Metabolic enzymes activity in alfalfa leaves.**   After feeding induction by adult alfalfa leaf weevils, the PAL enzyme activity of the alfalfa leaves was significantly higher than that of the healthy alfalfa treatment group ($P < 0.05$) (Fig 3A). As the feeding time increased, the PAL activity of all three treatments showed a trend of first increasing and then decreasing. The PAL of the one pest/plant and two pest/plant treatments reached their highest value after 60 h of feeding, which was 18.77% and 25.43% higher than that of the healthy alfalfa, respectively. The PAL of the three pest/plant treatment reached its highest level after 48 h of feeding induction, which was 21.53% higher than that of healthy alfalfa. As shown in Fig 3B, after feeding induction for 36, 48, 60, and 72 h, the PPO activity of the alfalfa leaves was significantly lower than that of healthy alfalfa ($P < 0.05$). The one pest/plant and two pest/plant treatments were the lowest after 60 h of feeding induction, which was 29.09% and 34.08% lower than that of healthy alfalfa, respectively. As shown in Fig 3C, the TAL activity of alfalfa leaves induced by feeding was significantly higher than that of healthy alfalfa ($P < 0.05$). The TAL activity of the one pest/plant and two pest/plant treatments was the highest at 60 h of feeding, which was 16.57% and 11.35% higher than that of healthy alfalfa, respectively. The three pest/plant treatments had the highest activity at 48 h of feeding, which was 18.07% higher than that of the healthy alfalfa. LOX activity induced by feeding on adult alfalfa leaf weevils was significantly higher than that of healthy alfalfa ($P < 0.05$) (Fig 3D). The LOX activity in alfalfa leaves was the highest in the one pest/plant and two pest/plant treatments after 60 h of feeding, which increased by 17.69% and 14.86%, respectively, compared to healthy alfalfa. The three pest/plant treatments had the highest activity after 48 h of feeding, which increased by 11.34% compared to that of healthy alfalfa.

**Protease inhibitors activity in alfalfa leaves.**   PIs are small molecular peptides and proteins that are widely distributed in different plant tissues. They can regulate endogenous protease activities in insects, participate in physiological processes, such as cell death, inhibit protease hydrolysis activities in the digestive tract of insects, and interfere with the normal growth and development of insects. The CI activity of alfalfa leaves induced by feeding on adult alfalfa leaf weevils was higher than that of healthy alfalfa leaves, with significant differences in CI activity at 36, 48, 60, and 72 h ($P < 0.05$) (Fig 4A). The data indicated that the CI activity of alfalfa leaves induced by the feeding of adult alfalfa leaf weevils showed a trend of first increasing and then decreasing with increasing feeding time. After 48 h of feeding, the CI activity of the one pest/plant and three pest/plant treatments was the highest, increasing by 28.34% and 28.73%, respectively, compared to that of healthy alfalfa. The two pest/plant treatments had the highest activity at 60 h of feeding, which was 23.50% higher than that of the healthy alfalfa. As shown in Fig 4B, the TI of alfalfa leaves induced by feeding on adult alfalfa leaf weevils was higher than that of healthy alfalfa. The TI activity of alfalfa leaves in the one

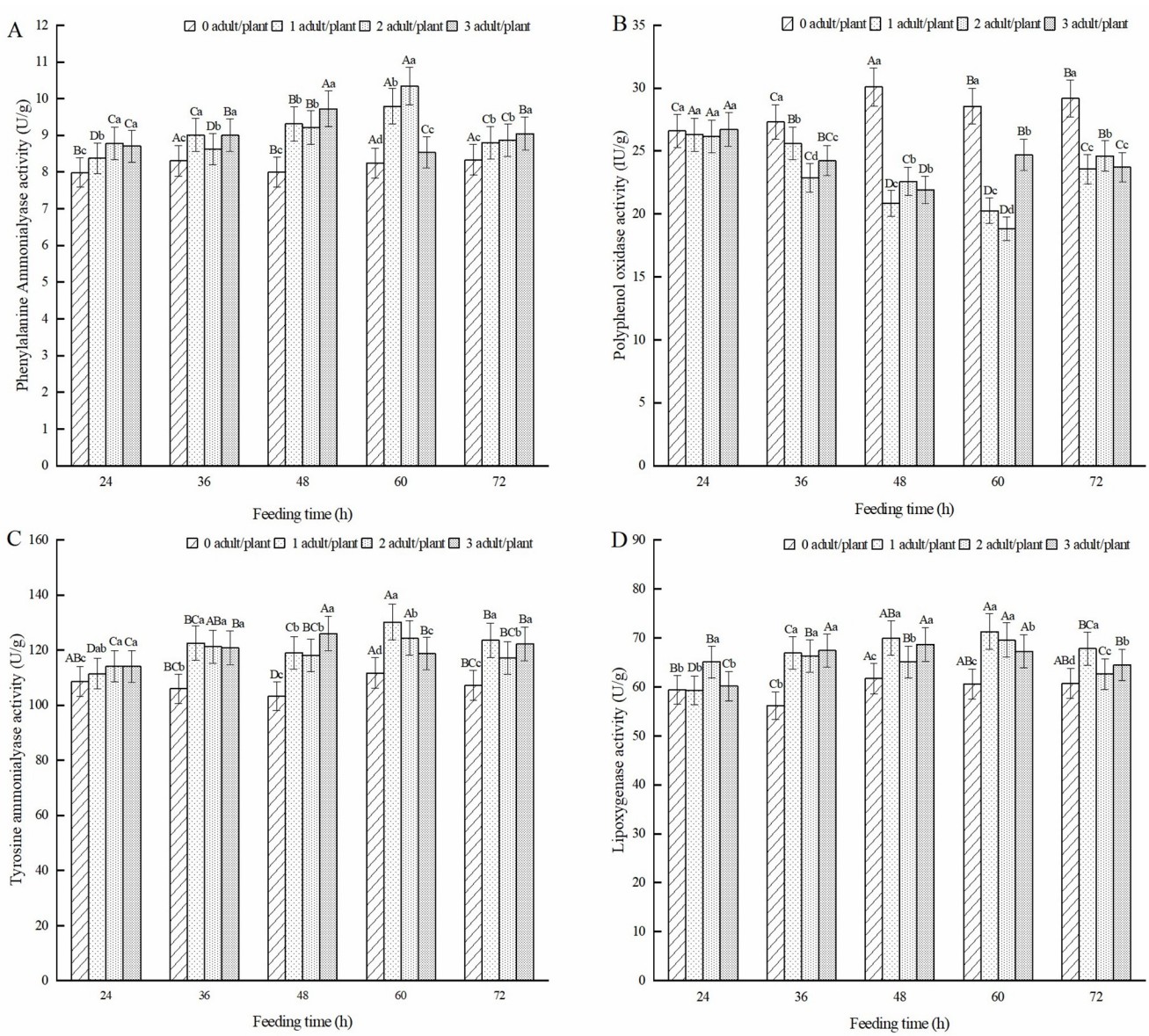

**Fig 3. Effect of metabolic enzymes in alfalfa leaves induced by adult alfalfa leaf weevils.** Different uppercase letters indicate significant differences in the same population density at different feeding times, whereas different lowercase letters indicate significant differences in population density at the same feeding time ($P < 0.05$, Duncan multiple comparisons).

pest/plant, and two pest/plant treatments showed alternating increases and decreases with an increase in feeding time in alfalfa leaf weevil adults, and both showed the highest TI activity at 60 h of feeding, which was 20.31% and 12.75% higher than that of healthy alfalfa, respectively. The three pest/plant treatments showed decreasing and increasing trends with increasing feeding time, and the highest TI activity was observed in alfalfa leaves after 48 h of feeding, which was 16.21% higher than that of healthy alfalfa.

**JA content in alfalfa leaves.** JA plays a key role in plant defense against herbivorous insects. JA regulates plant resistance to herbivorous insects and necrotrophic pathogens, including nematodes, chewing and sucking insect pests. Studies have found that plants rapidly

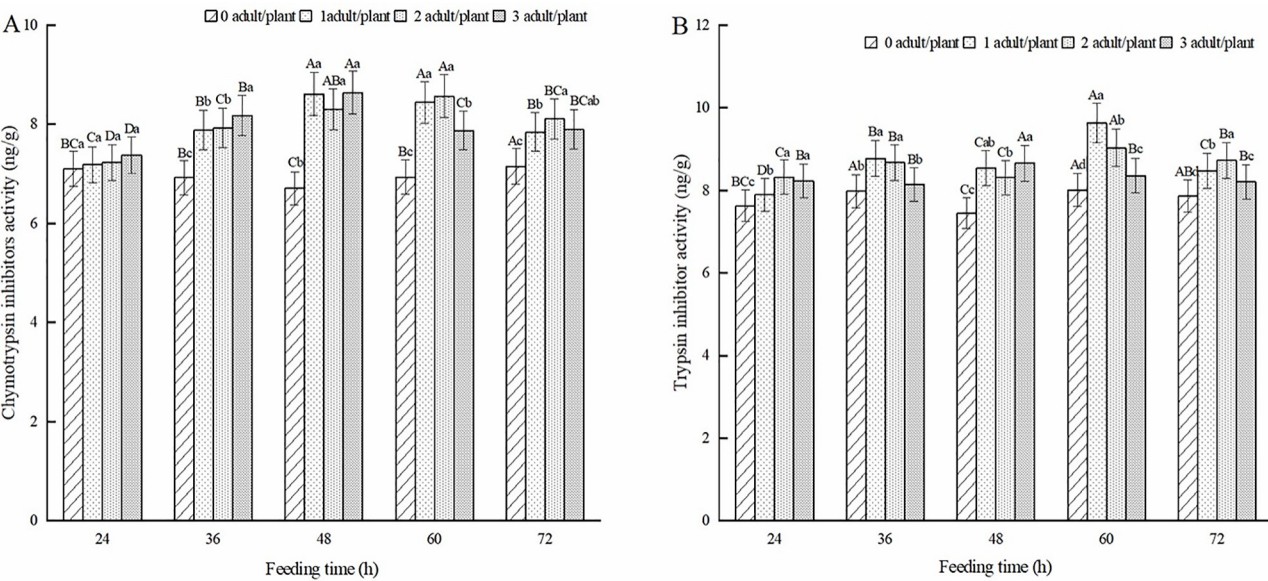

**Fig 4. Effect of protease inhibitors in alfalfa leaves induced by adult alfalfa leaf weevils.** Different uppercase letters indicate significant differences in the same population density at different feeding times, whereas different lowercase letters indicate significant differences in population density at the same feeding time ($P < 0.05$, Duncan multiple comparisons).

accumulate JA after being subjected to biological stress, and genes related to JA biosynthesis are significantly upregulated. As shown in Fig 5, after feeding on adult alfalfa leaf weevils, the JA content in alfalfa leaves was significantly higher than that in healthy alfalfa leaves with increasing feeding time ($P < 0.05$). The JA content in the one pest/plant treatment showed a slow upward trend and reached its maximum value at 72 h of feeding, an increase of 18.37% compared with healthy alfalfa. The two pest/plant treatments showed a trend of first increasing, then decreasing, and then increasing again, reaching a maximum value at 72 h of feeding. The three pest/plant treatments showed a trend of first increasing and then decreasing, reaching a maximum value at 48 h of feeding, which increased by 27.30% compared to healthy alfalfa.

**Effects of adult density and feeding time of alfalfa leaf weevils on related indicators.** Pearson's correlation analysis was used to study the relationships between these indicators (Fig 6). The density of adult alfalfa leaf weevils was positively correlated with PAL, MDA, LOX, CAT, TAL, TI, T-AOC, and JA content. The correlation coefficients of density of adult alfalfa leaf weevils with MDA, TAL, and T-AOC were 0.57, 0.56, and 0.50, respectively, reached a significant level ($P < 0.05$). The density of adult alfalfa leaf weevils was negatively correlated with SOD, PPO, POD, and CI levels. The correlation coefficients of the density of adult alfalfa leaf weevils with PPO and POD were −0.48 and −0.47, respectively, reaching a significant level ($P < 0.05$). The feeding time of alfalfa leaf weevils was positively correlated with PAL, MDA, LOX, SOD, TAL, TI, CI, T-AOC, and JA content. The correlation coefficient of feeding time with CI was 0.65, reaching a significant level ($P < 0.05$). The feeding time of alfalfa leaf weevils was negatively correlated with CAT and POD but not significantly.

We selected two principal components whose cumulative contribution rate of the eigenvalue reached 73.5% (Fig 7A). The variance contribution rates of principal components 1 and 2 were 55.1% and 18.4%, respectively, indicating that they effectively reflected the original data in the auxiliary indicators. The loading plot for the principal components was used to measure

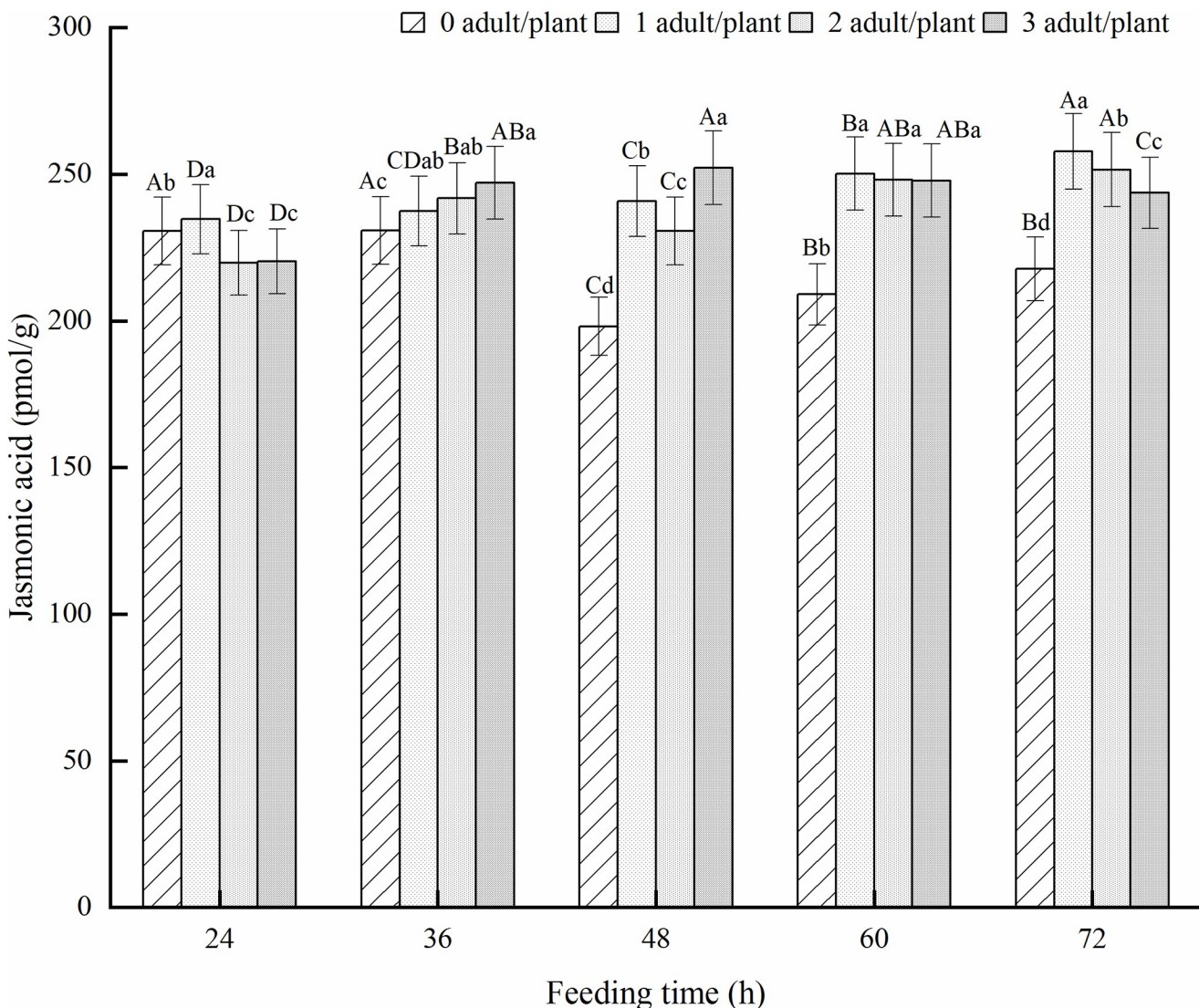

**Fig 5. Effect of JA in alfalfa leaves induced by adult alfalfa leaf weevils.** Different uppercase letters indicate significant differences in the same population density at different feeding times, whereas different lowercase letters indicate significant differences in population density at the same feeding time (*P* < 0.05, Duncan multiple comparisons).

their contributions. Specifically, a larger absolute value of the load implies that the contribution of the corresponding principal component is larger. Principal component 1 had a large to small load in terms of LOX, POD, PPO, TAL, MDA, T-AOC, TI, PAL, and JA (Fig 7B). Principal component 2 showed a large load in terms of SOD, CI, and CAT (Fig 7B). These results showed that, except for SOD, CI, and CAT, other indicators could reflect the effect of feeding induction of alfalfa leaf weevils on alfalfa to a large extent, especially LOX, POD, PPO, TAL, MDA, T-AOC, TI, PAL, and JA.

### Effect of defensive substances in alfalfa leaves induced by 3rd-instar larvae of alfalfa leaf weevils

**Antioxidant enzymes activity in alfalfa leaves.** The effect of feeding induction by alfalfa leaf weevil larvae on defensive substances in the alfalfa leaves is shown in Fig 8. As shown in

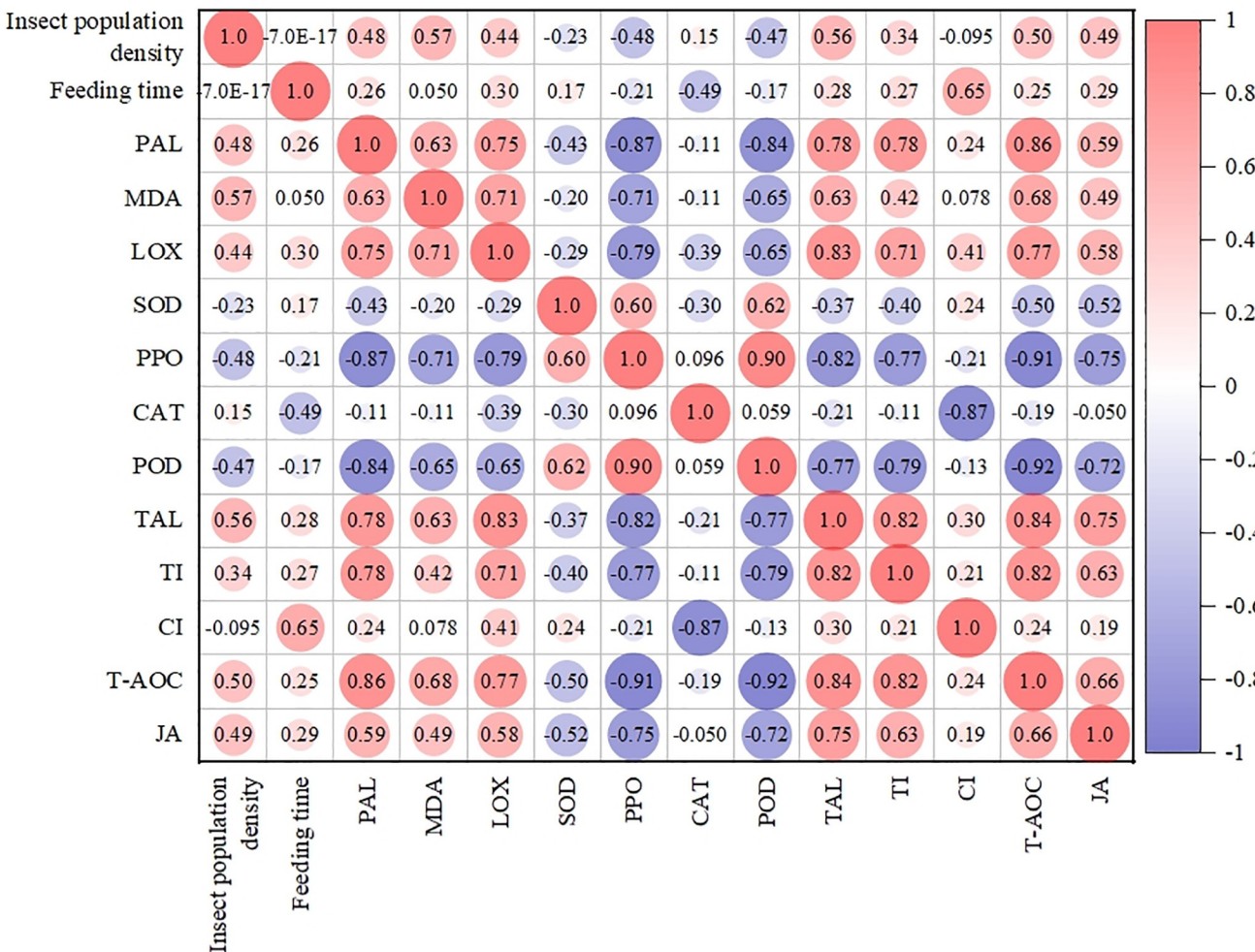

**Fig 6. Heat map of Pearson correlation coefficients.** The color and size of the circles represent pearson correlation coefficient and p values, respectively. The deep and large circles indicate significant correlations.

Fig 8A, after feeding induction by alfalfa leaf weevil larvae, the CAT activity of alfalfa leaves was significantly lower than that of the healthy alfalfa ($P < 0.05$). The CAT activity of alfalfa leaves after treatment with two pests/plant and four pests/plant showed an overall trend of first increasing and then decreasing, with the highest activity occurring after 36 h of feeding. The CAT activity of the six pest/plant treatment groups was highest after 24 h of feeding. It is worth noting that the CAT activity of all three treatments was the lowest at 72 h of feeding, which was 29.3%, 30.1%, and 27.5% lower than that of healthy alfalfa. The POD activity of alfalfa leaves induced by feeding on alfalfa leaf weevil larvae was significantly lower than that of the healthy alfalfa ($P < 0.05$) (Fig 8B). As the feeding induction time increased, the POD activity showed a decreasing trend, with the lowest POD activity in alfalfa leaves after 72 h of feeding, which was 41.2%, 35.3%, and 41.0% lower than that of healthy alfalfa. As shown in Fig 8C, after feeding alfalfa leaf weevil larvae, the SOD activity of alfalfa leaves was higher than that of healthy alfalfa at 24 h and significantly lower than that of healthy alfalfa at 36 h, 48 h, 60 h, and 72 h ($P < 0.05$). Under the same pest density, the SOD activity in alfalfa leaves showed a decreasing trend with increased feeding time, and the lowest activity was observed after 72 h of

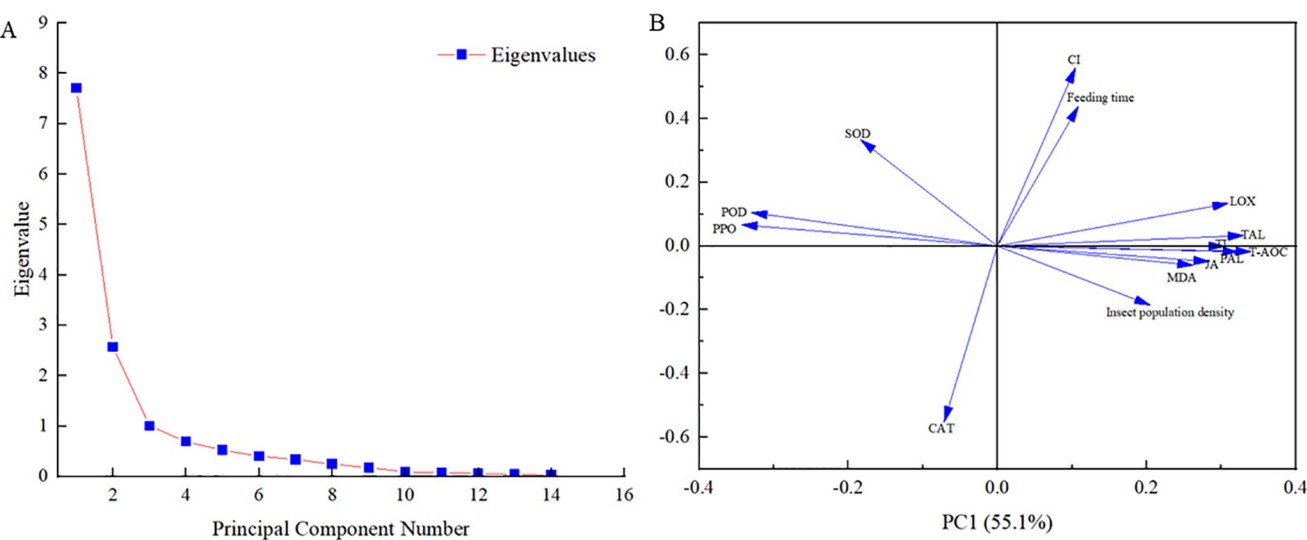

**Fig 7. Scree plot of eigenvalues for principal components (A) and loading plots for principal components 1 and 2 (B).**

feeding. Compared to healthy alfalfa, the SOD activity decreased by 22.8%, 29.9%, and 32.9%. As shown in Fig 8D and 8E, the content of T-AOC and MDA in alfalfa leaves induced by feeding of alfalfa leaf weevil larvae was significantly higher than that of healthy alfalfa ($P < 0.05$). The T-AOC of alfalfa leaves treated with 2 and 6 pests/plant reached its highest value at 60 h, which was 28.3% and 27.6% higher than that of healthy alfalfa. Four pests/plants reached their highest value at 72 h, an increase of 43.0% compared with healthy alfalfa (Fig 8D). The MDA content of each treatment group reached its highest value after 72 h of feeding, which increased by 15.7%, 18.6%, and 17.0 compared to healthy alfalfa (Fig 8E).

**Metabolic enzymes activity in alfalfa leaves.** As shown in Fig 9A, the PAL activity of alfalfa leaves induced by feeding on alfalfa leaf weevil larvae was significantly higher than that of the healthy alfalfa ($P < 0.05$). The PAL activity of alfalfa leaves in the four pest/plant and six pest/plant treatments increased with increasing feeding time, reaching the highest level after 72 h of feeding. Compared to healthy alfalfa, the PAL activity increased by 10.6% and 19.0%. The PAL activity of the alfalfa leaves in the two pest/plant treatments showed an overall trend of first increasing and then decreasing, with the highest PAL activity observed after 48 h of feeding. The PPO activity of alfalfa leaves induced by feeding on alfalfa leaf weevil larvae was significantly lower than that of the healthy alfalfa ($P < 0.05$) (Fig 9B). As the feeding time increased, the PPO activity of alfalfa leaves decreased. The PPO activity of the two pest/plant and six pest/plant treatment groups was the lowest at 72 h of feeding, which was 47.2% and 48.3% lower than that of the healthy alfalfa, respectively. The TAL activity of alfalfa leaves induced by feeding on alfalfa leaf weevil larvae is shown in Fig 9C. The TAL activity in alfalfa leaves showed a trend of first increasing, then decreasing, and then increasing with increasing feeding time in the two pest/plant and four pest/plant treatments. The TAL activity of the six pest/plant treatment groups increased with increasing feeding time. The TAL activity of the three treatments was highest after 72 h of feeding, with increases of 14.7%, 11.4%, and 10.7% compared to that of healthy alfalfa. As shown in Fig 9D, the LOX activity induced by feeding on alfalfa leaf weevil larvae was significantly higher than that in the control group (P < 0.05). As the feeding time increased, the LOX activity of alfalfa leaves showed a trend of first increasing and then decreasing, with the highest activity observed after 60 h of feeding, which was 30.8%, 27.4%, and 29.3% higher than that of healthy alfalfa.

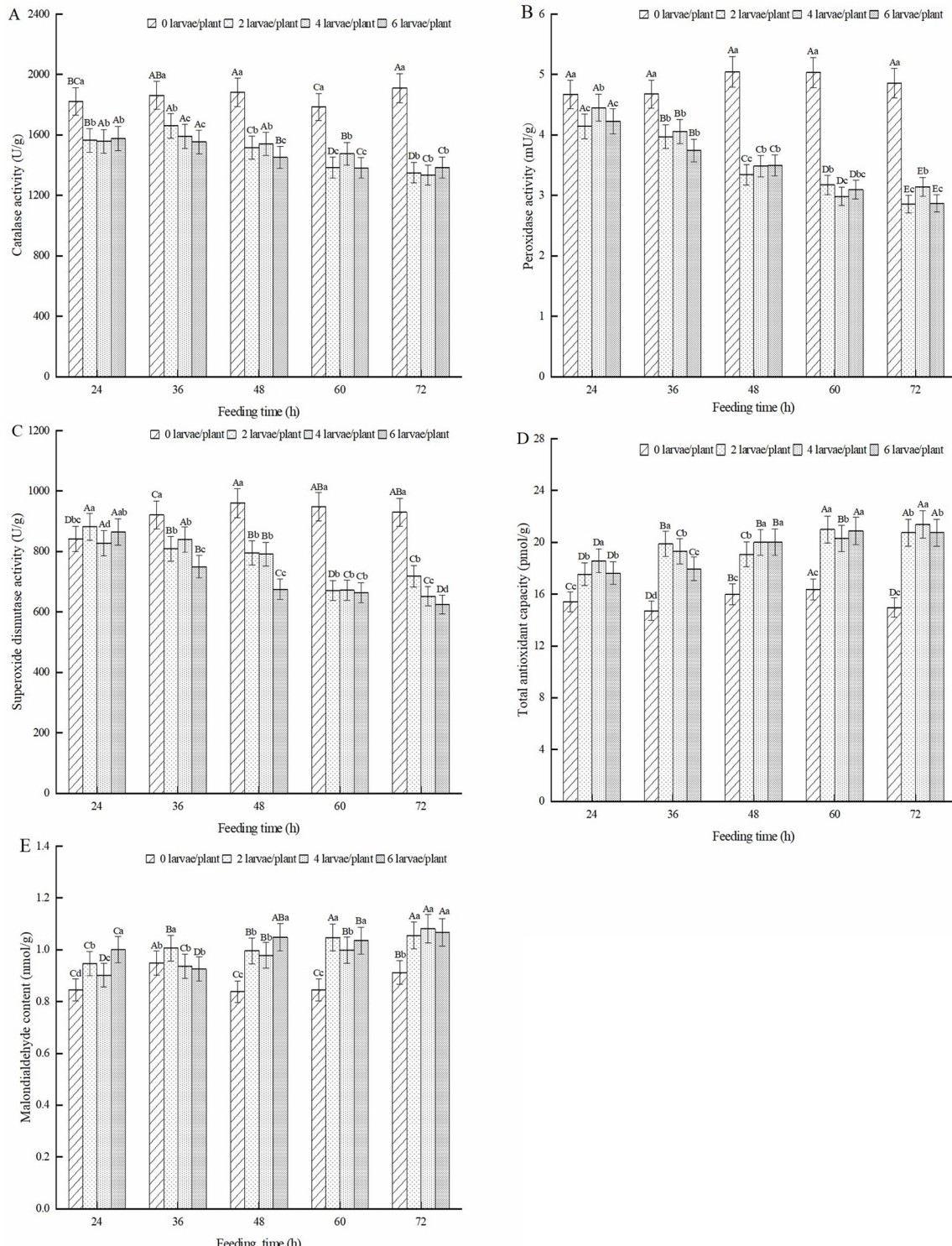

**Fig 8. Effect of antioxidant enzymes in alfalfa leaves induced by 3rd-instar larvae of alfalfa leaf weevils.** Note: Different uppercase letters indicate significant differences in the same population density at different feeding times, whereas different lowercase letters indicate significant differences in population density at the same feeding time ($P < 0.05$, Duncan multiple comparisons).

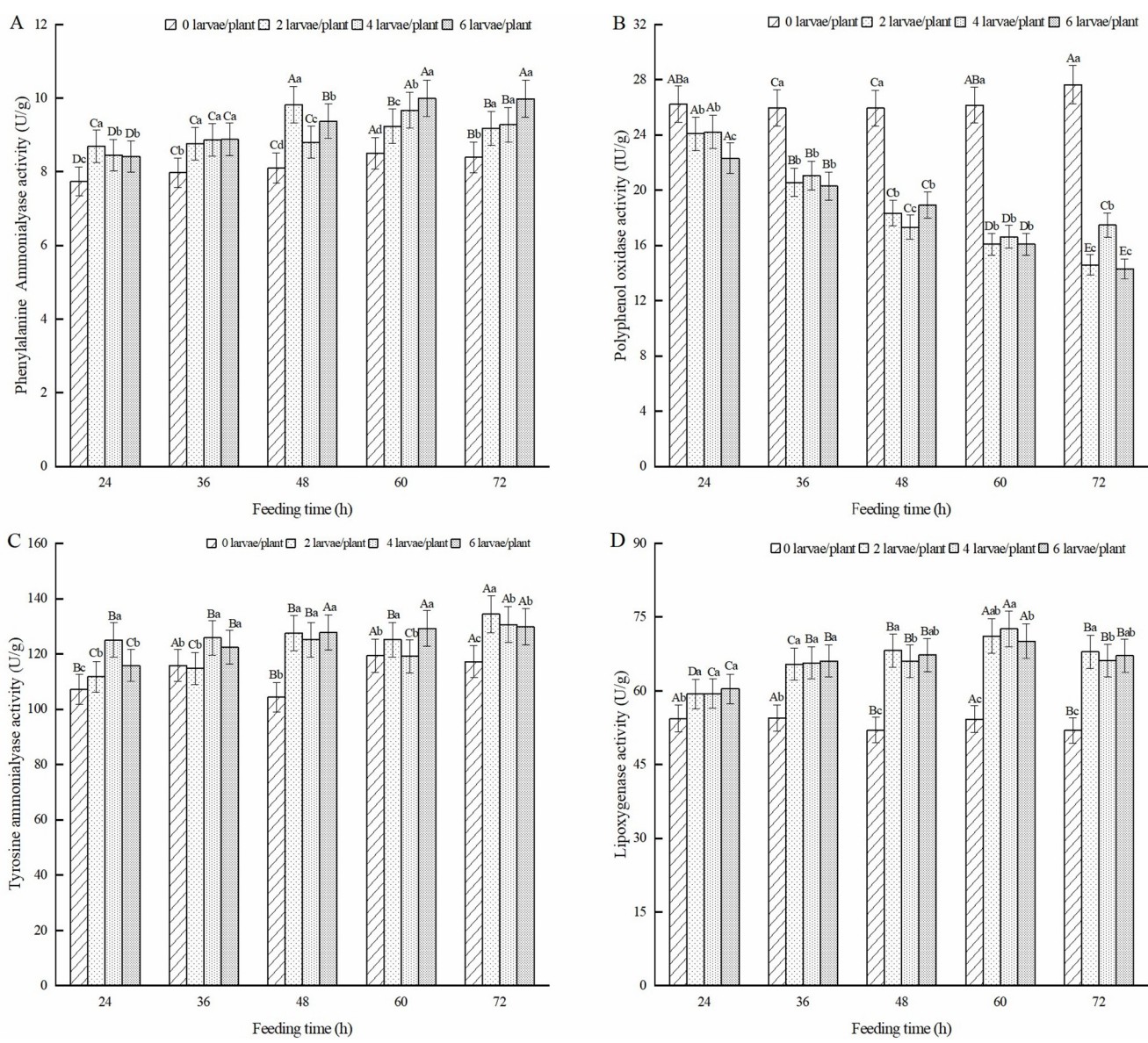

**Fig 9. Effect of metabolic enzymes in alfalfa leaves induced by 3rd-instar larvae of alfalfa leaf weevils.** Different uppercase letters indicate significant differences in the same population density at different feeding times, whereas different lowercase letters indicate significant differences in population density at the same feeding time ($P < 0.05$, Duncan multiple comparisons).

**Protease inhibitors activity in alfalfa leaves.** As shown in Fig 10A, the CI activity of alfalfa leaves induced by feeding on alfalfa leaf weevil larvae was significantly higher than that of the healthy alfalfa ($P < 0.05$). The group treated with four pests/plant had the highest CI activity in alfalfa leaves after 60 h of feeding, which was 33.9% higher than that of healthy alfalfa. The CI activity of alfalfa leaves was the highest in both the two pest/plant and six pest/plant treatments after 72 h of feeding, which increased by 31.9% and 29.5% compared to healthy alfalfa, respectively. As shown in Fig 10B, the TI content of alfalfa leaves in the larval feeding treatment was significantly higher than that of healthy alfalfa ($P < 0.05$). With an increase in feeding time, the TI content of alfalfa leaves induced by feeding showed an upward trend, reaching its maximum value after 72 h of feeding, which was 25.9%, 30.7%, and 26.1% higher than that of healthy alfalfa.

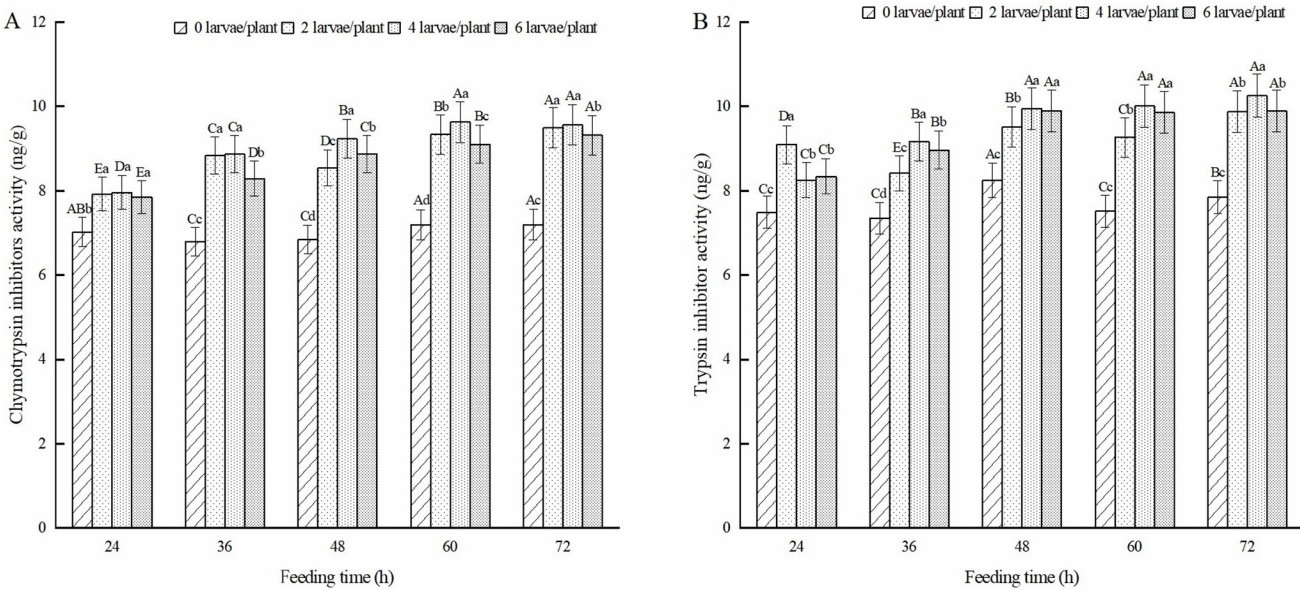

**Fig 10. Effect of protease inhibitors in alfalfa leaves induced by 3rd-instar larvae of alfalfa leaf weevils.** Different uppercase letters indicate significant differences in the same population density at different feeding times, whereas different lowercase letters indicate significant differences in population density at the same feeding time ($P < 0.05$, Duncan multiple comparisons).

**JA content in alfalfa leaves.** The JA content in the alfalfa leaves of the treatment group fed with alfalfa leaf weevil larvae was significantly higher than that of healthy alfalfa ($P < 0.05$) (Fig 11). The JA content in the leaves of alfalfa treated with two pests/plant showed an upward trend and reached its maximum value after 72 h of feeding, which was 30.1% higher than that of healthy alfalfa. The JA content in the alfalfa leaves of the four pest/plant and six pest/plant treatments showed an alternating trend of increase and decrease, reaching their maximum values at 60 and 72 h of feeding, respectively.

**Effects of larvae density and feeding time of alfalfa leaf weevils on related indicators.** Pearson's correlation analysis was used to study the relationships between these indicators (Fig 12). The density and feeding time of alfalfa leaf weevil larvae were significantly positively correlated with PAL, MDA, LOX, TAL, TI, CI, T-AOC, and JA content ($P < 0.05$). The density and feeding time of alfalfa leaf weevil larvae were negatively correlated with SOD, PPO, CAT, and POD activities ($P < 0.05$).

We selected two principal components whose cumulative contribution rate of the eigenvalue reached 86.3% (Fig 13A). The variance contribution rates of principal components 1 and 2 were 78.6% and 7.7%, respectively, indicating that they effectively reflected the original data in the auxiliary indicators. Principal component 1 had a large to small load in terms of TAL, CAT, LOX, PAL, T-AOC, MDA, SOD, TI, CI, POD, PPO, and JA (Fig 13B). These results showed that all indicators could reflect the effect of feeding induction by 3rd-instar larvae of alfalfa leaf weevils on alfalfa to a large extent.

## Discussion

### Effects of alfalfa leaf weevil feeding on antioxidant enzyme activity in alfalfa

When pests feed on and invade plants, they induce changes in defense enzyme activities to resist the stress of pests [26]. The mechanisms of defense enzymes, such as POD, PAL, PPO,

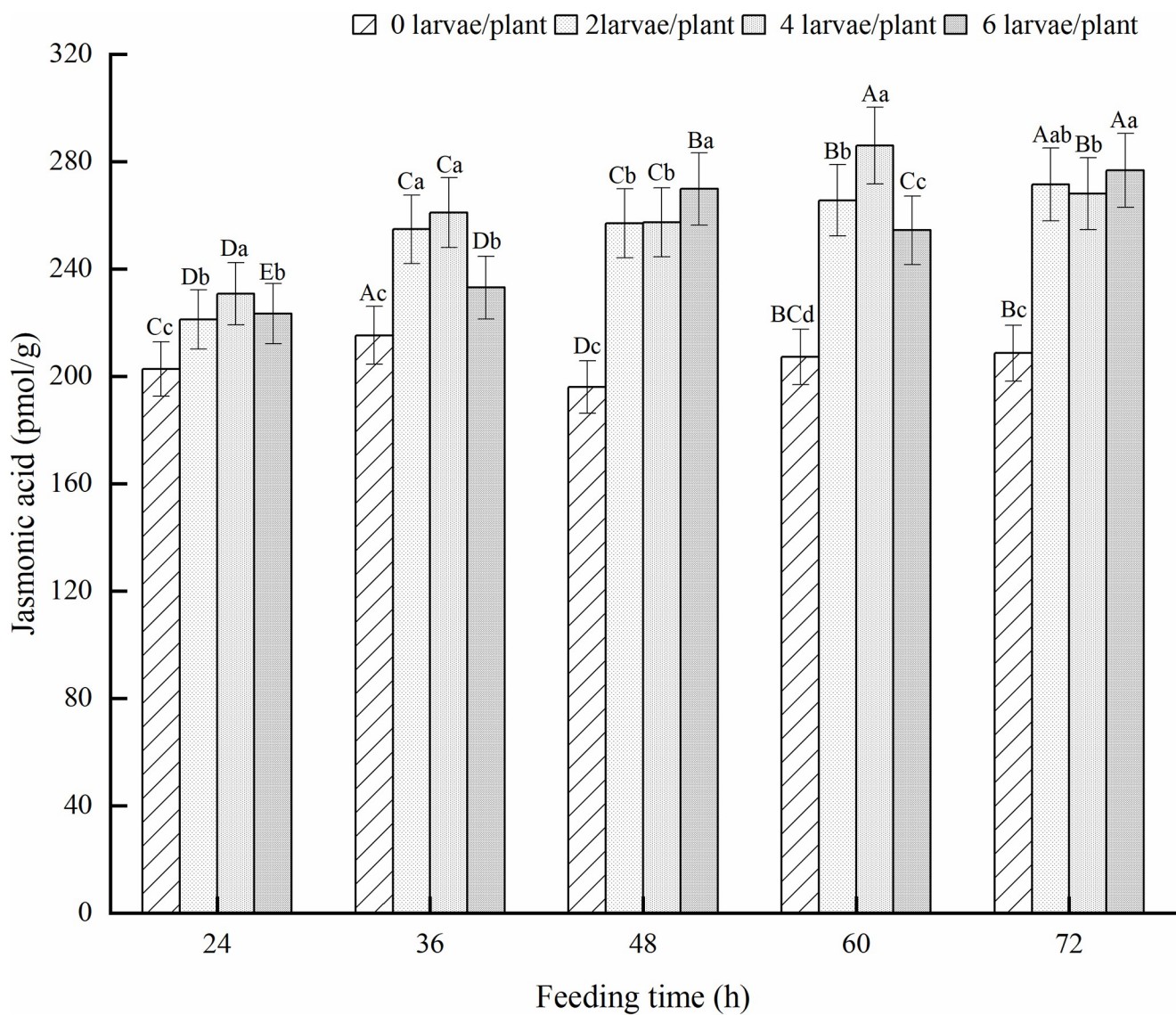

**Fig 11. Effect of JA in alfalfa leaves induced by 3rd-instar larvae of alfalfa leaf weevils.** Different uppercase letters indicate significant differences in the same population density at different feeding times, whereas different lowercase letters indicate significant differences in population density at the same feeding time ($P < 0.05$, Duncan multiple comparisons).

CAT, and SOD, in plants have both similarities and obvious differences. They play an important role in plant responses to pest stress through their coordinated action [27]. Insect feeding can produce large number of ROS toxic substances in plants [13]. To cope with phytophagous insects, plants produce substances such as POD, CAT, SOD, and MDA and maintain the balance of ROS in the body by directly or indirectly removing excess oxidative free radicals in plants [18]. Wu found that the POD activity of alfalfa leaves was lower than that of CK when the thrips density was nine pest/plant [28]. The activities of CAT, SOD, and PPO in alfalfa leaves of both resistant and susceptible thrips alfalfa varieties were lower than those of CK at a density of 5–9 pest/plant. Our study showed that the POD, CAT, and SOD of alfalfa leaves were lower than those of healthy alfalfa after feeding induction of adults and larvae of alfalfa

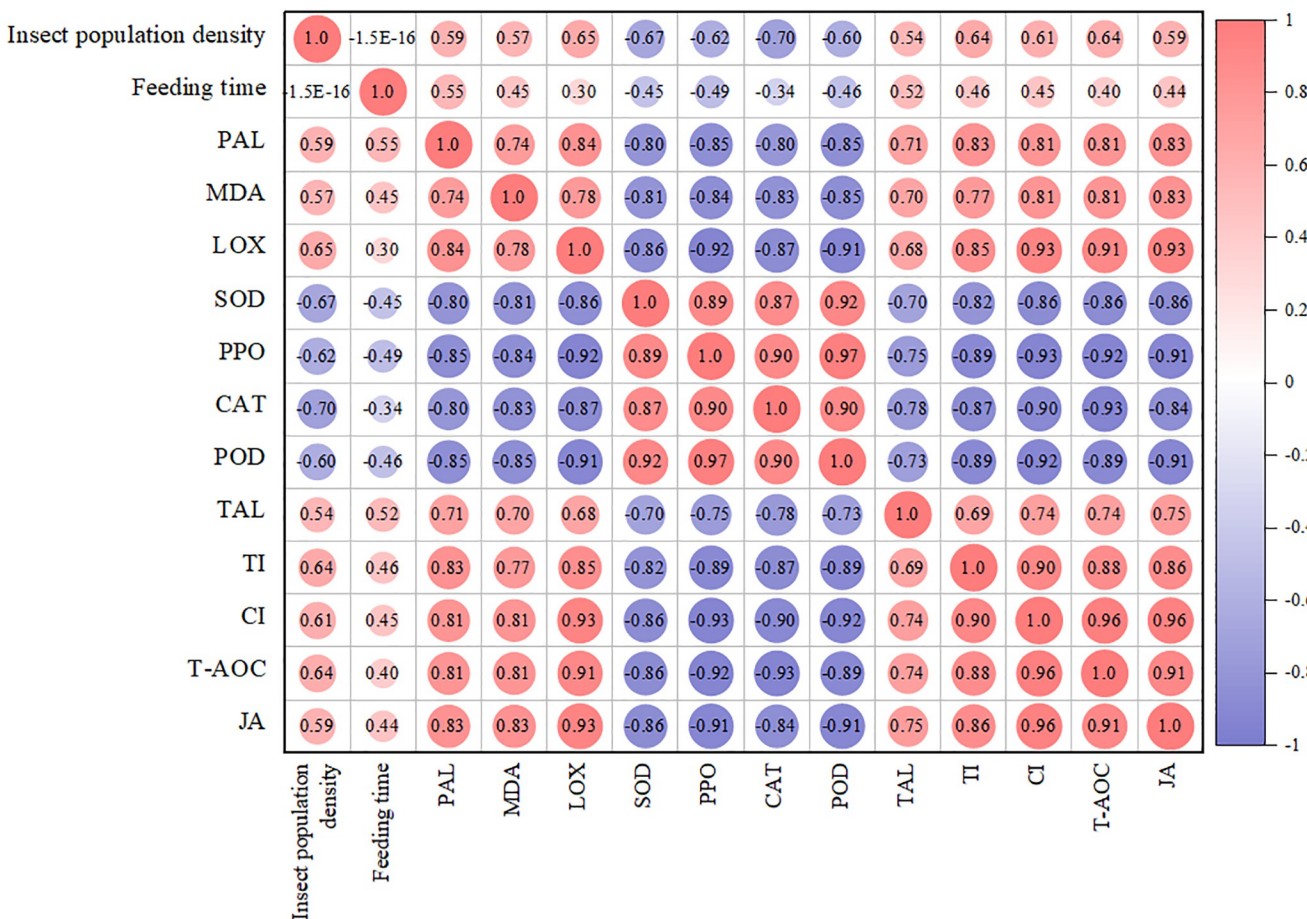

**Fig 12. Heat map of Pearson correlation coefficients.** The color and size of the circles represent pearson correlation coefficient and *p* values, respectively. The deep and large circles indicate significant correlations.

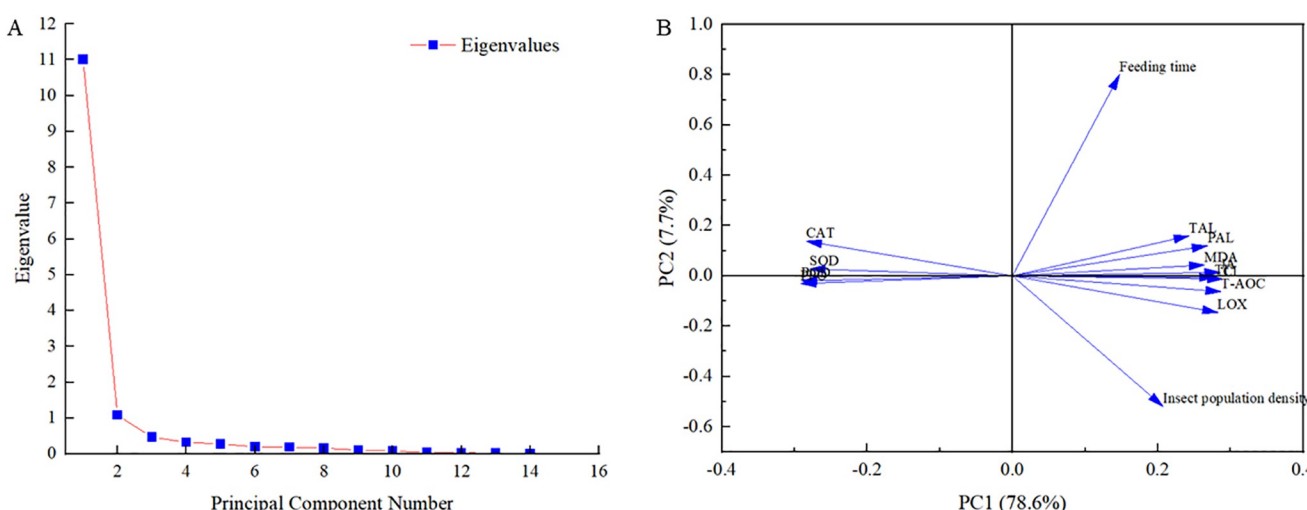

**Fig 13. Scree plot of eigenvalues for principal components (A) and loading plots for principal components 1 and 2 (B).**

leaf weevils, showing a trend of first decreasing and then increasing. It showed that POD, CAT, and SOD played major roles in ROS production by insect feeding stress in alfalfa. At the late stage of feeding induction by adults of alfalfa leaf weevil, the activities of POD, CAT, and SOD in alfalfa leaves increased but were still lower than those in the early stage of feeding induction. This is consistent with the change in the defense enzyme activity of thrips fed alfalfa leaves [27]. At the initial feeding period (24 h), the activity of CAT and PPO feeding by adults of alfalfa leaf weevils showed no significant change compared to healthy alfalfa but gradually decreased after that, which was lower than that of healthy alfalfa. However, after feeding induction by alfalfa leaf weevil larvae, the CAT and PPO activities were lower than those of healthy alfalfa. Similarly, the POD and SOD activity in alfalfa leaves decreased rapidly after feeding alfalfa leaf weevil larvae, which was significantly lower than that of healthy alfalfa.

In this study, the T-AOC and MDA contents in the leaves of alfalfa plants induced by feeding were significantly higher than those of healthy alfalfa. This indicated that the accumulation of ROS caused by alfalfa leaf weevil feeding stimulated the scavenging ability of the ROS system in alfalfa. With the extension of the feeding time of alfalfa leaf weevils, T-AOC increased at first and then decreased, which also reflected that alfalfa plants could not remove excess ROS timely and effectively because of the feeding stress of heavy insect population density for a long time, so they could not resist the injury caused by ROS and the insects continuous feeding [29]. MDA is the product of membrane lipid peroxidation under plant stress, which can reduce the content of unsaturated fatty acids in the membrane, change the function and structure of the membrane, and ultimately lead to the loss of membrane structure and physiological functions [30]. We found that both larval and adult feeding induction of alfalfa leaf weevils could cause the content of MDA in alfalfa leaves to increase, and it showed an upward trend with the increase in feeding time, indicating that the cell membrane system of alfalfa was damaged under the stress of insect feeding.

## Effect of alfalfa leaf weevil feeding on defense metabolic enzymes in alfalfa

Plants not only produce antioxidant enzymes to eliminate ROS caused by pest feeding but also induce enzyme activity related to defense pathways, actively attacking to achieve the effect of insect resistance. PAL is a key rate-limiting enzyme in plants that connects the primary metabolism oxalic acid pathway and the secondary metabolism phenylpropanoid pathway. It plays an important role in the synthesis of many secondary defense-related compounds, such as phenols, lignin, and phytoprotectants. Research has shown that *Lymantria dispar* feeds on poplar trees, thrips feeds on alfalfa, *F. occidentalis* feeds on tomatoes, *Empoasca onukii* feeds on tea trees, and *Therioaphis trifolii* feeds on alfalfa caused changes in PAL activity and its gene expression in host plants [27,31–34]. We found that the PAL activity of alfalfa leaves induced by feeding on adult and larval alfalfa leaf weevils was higher than that of healthy alfalfa and showed a trend of first increasing and then decreasing with increasing feeding time.

PPO can oxidize phenolic substances in plants into highly toxic quinones, which affect insect development. It not only hinders the utilization of plant nutrients by insects but also affects their growth and development until death [35]. We found that after feeding induction by adult and larval alfalfa leaf weevils, the PPO activity in alfalfa leaves was lower than that in healthy alfalfa. With an increase in feeding time, the PPO content showed a first decreasing and then increasing trend, while the larval treatment showed a continuous downward trend. This result was consistent with the PPO activity of *T. trifolii* and thrips feeding on different alfalfa varieties [27,31]. This indicates that alfalfa leaves exhibit a faster response to the feeding of adult alfalfa leaf weevils and that the highly toxic quinone substances formed by PPO oxidation have a positive effect on the feeding resistance of adults. Adults of alfalfa leaf weevils feed

on alfalfa stems and leaves, whereas larvae feed on leaf flesh and leaf veins [36]. PPO is distributed in flower organs, meristems, leaves, stems, and roots; therefore, the feeding habits of adults promote the active defense of alfalfa leaves to resist invasion.

LOX is a key enzyme in the JA-mediated octadecanoic acid pathway in plants, catalyzing the oxygenation reaction of unsaturated fatty acids to form JA and plays an important role in plant stress signal recognition and transduction. Pu et al. believed that after F. occidentalis feeding, the LOX activity in tomatoes increased. In this study, after feeding on adult and larval alfalfa leaf weevils, the LOX activity in alfalfa leaves was higher than that of healthy alfalfa. With increasing feeding time, the LOX activity showed a trend of first increasing and then decreasing, reaching its highest value after 60 h of feeding. TAL is a key enzyme in the synthesis of phenolic substances, and its increased activity is conducive to the formation of phenolic substances, whereas PPO is a key enzyme in the phenolic metabolic system. He et al. found that after feeding on *Paspalum dilatatum*, the TAL activity of *Oxya chinensis* showed a trend of first increasing and then decreasing with the prolongation of feeding time, reaching peaks at 48, 72, and 24 h [37]. Our results indicated that after feeding on alfalfa leaf weevil larvae and adults, the overall TAL content in alfalfa leaves showed an upward trend.

## Effect of alfalfa leaf weevil feeding on alfalfa protease inhibitors

Many plants synthesize PIs after being fed by insects. PIs can inhibit digestive enzymes in pests at a high level and are important natural defense measure for plants to resist pests [38]. TIs and CIs could bind to insect intestinal proteolytic enzymes, inhibit insect digestive enzyme activity, disrupt normal insect metabolism, affect normal insect development, and even lead to death [39]. Our study found that after feeding on alfalfa leaf weevil larvae and adults, the TI and CI activities of alfalfa leaves were significantly higher than those of healthy alfalfa. This indicates that alfalfa produces PI to defend against insect damage when subjected to feeding stress by alfalfa leaf weevils. This is consistent with the significant increase in TI and CI activities induced by feeding on *Dendrolimus superans* [40]. After feeding induction by adult alfalfa leaf weevils, both TI and CI activities in alfalfa leaves showed a trend of first increasing and then decreasing, while TI and CI showed an overall upward trend under feeding by alfalfa leaf weevil larvae, which is closely related to the different feeding habits of different insect states.

## Conclusions

In summary, feeding induction of alfalfa leaf weevil larvae and adults could trigger changes in defense-related enzymes in alfalfa. After feeding induction by adults and larvae of alfalfa leaf weevils, T-AOC, MDA, PAL, TAL, LOX, CI, TI, and JA in the alfalfa leaves increased with increasing feeding time. Alfalfa resists the harm caused by alfalfa leaf weevils by increasing the enzymes and related hormones required for defense. After feeding induction by adult alfalfa leaf weevils, the activities of CAT, POD, SOD, and PPO in alfalfa leaves first increased and then decreased, however, their activities showed a downward trend, indicating differences in the feeding-induced defense responses to the two life feeding stages of alfalfa leaf weevils. It was due to different feeding habits of adults and larvae of alfalfa leaf weevils. This study clarified the dynamic relationship between alfalfa leaf weevil feeding and defensive substances in the alfalfa. This provides theoretical support for research on insect resistance mechanisms and alfalfa breeding.

## Supporting information

**S1 File. Method for determining enzyme activity indicators.**
(DOCX)

## Author Contributions

**Conceptualization:** Chunhui Ma.

**Data curation:** Hui Liu.

**Funding acquisition:** Chunhui Ma.

**Investigation:** Hui Liu, Xuzhe Wang, Yong Ma, Wanshun Gao.

**Methodology:** Hui Liu.

**Project administration:** Hui Liu, Xuzhe Wang, Yong Ma, Wanshun Gao.

**Supervision:** Chunhui Ma.

**Writing – original draft:** Hui Liu.

**Writing – review & editing:** Hui Liu, Xuzhe Wang, Chunhui Ma.

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
