## [Decision Letter · Decision Letter 0]

9 Aug 2024

PONE-D-24-23169Alfalfa Leaf Weevil Larvae and Adults Feeding Induces Physiological Response in AlfalfaPLOS ONE

Dear Dr. Liu,

Thank you for submitting your manuscript to PLOS ONE. After careful consideration, we feel that it has merit but does not fully meet PLOS ONE’s publication criteria as it currently stands. Therefore, we invite you to submit a revised version of the manuscript that addresses the points raised during the review process.

The manuscript has been reviewed by experts and on the basis of their comments and decision the manuscript needs revision. Editor and reviewer comments are included below.

A marked-up copy of your manuscript that highlights changes made to the original version. You should upload this as a separate file labeled 'Revised Manuscript with Track Changes'.An unmarked version of your revised paper without tracked changes. You should upload this as a separate file labeled 'Manuscript'.

We look forward to receiving your revised manuscript.

Kind regards,

S Ezhil Vendan, Ph.D

Academic Editor

PLOS ONE

Journal Requirements:

2. Thank you for stating the following financial disclosure: "  National Modern Agriculture Industry Technology System (CARS-34) of the Ministry of Agriculture and Rural Affairs".

Additional Editor Comments:

The manuscript described about the feeding activity of alfalfa leaf weevil larvae and adults, and physiological changes in alfalfa. The study design and manuscript preparation was well. Study background information, objective and methodology was clearly described in the manuscript. However, results and interpretations sections need revision. Authors needs to consider the following comments to improve the manuscript.

Line 29: In addition to cow, add two more examples.

Line 105: Write expansion of PBS.

Line 426: Remove “[34]”

Methodology for each enzyme assay may describe in the methodology section instead of providing as supplementary section. Supportive tables/figures/details may provide as supplementary information.

Add methodology for Principal Component analysis.

In section 3.1. and 3.2.: Avoid repetitions of “in alfalfa leaves induced by adult alfalfa leaf weevils” and “in alfalfa leaves induced by 3rd-instar larvae of alfalfa leaf Weevils” in subsection titles.

In Figure 2 - 5: Remove abbreviations in Y axis titles.

In Figure 6 – 7 & 12 - 13: Write expansions for all the abbreviations in the figure title.

In Figures: Results are expressed in enzyme specific activity based units. Whereas in the text, results were described in percentage levels. Please check and describe clearly with respect to the presented results or add additional informations for better understanding.

Please check the cited references with the reference list vice versa. Check and write all the references as per the journal guidelines. For example, see line 507-508, 526-527, 546, 547, etc.,

Please check the typographical errors throughout the manuscript.

Reviewers' comments:

Reviewer's Responses to Questions

**Comments to the Author**

1. Is the manuscript technically sound, and do the data support the conclusions?

Reviewer #1: Yes

Reviewer #2: Yes

2. Has the statistical analysis been performed appropriately and rigorously? 

Reviewer #1: Yes

Reviewer #2: Yes

3. Have the authors made all data underlying the findings in their manuscript fully available?

Reviewer #1: Yes

Reviewer #2: Yes

4. Is the manuscript presented in an intelligible fashion and written in standard English?

Reviewer #1: Yes

Reviewer #2: Yes

5. Review Comments to the Author

Reviewer #1: Cohesive, beautifully structured MS. The choice of topic is also an economically important field, but the stress physiology of insect damage to alfalfa is also important in basic research. The methods and implementation of the experimental design and research work are of scientific standard. The text of the thesis is easy to follow, its structure is based on a nice internal logic. Perhaps some of the groundwork ideas from Discussion could have been included in Introduction. In evaluating the results, correlations are sought and found. Unfortunately, many sentences in the conclusions chapters are identical to some sentences in the abstract and discussion chapters. Please check this for the other quote species as well.

Other comments:

ad 89-90: Do I need a company name? Hoagland solution is well-known, easy to prepare, widely used.

ad 160 & 284: Proposed simplification: metabolic enzymes

ad 190: Is this citation really relevant ?

ad 198: TI or TPI?

AD 367/375/412/415: It is not customary to put the species descriptive auctor in parentheses. It is enough to specify the auctor at the first summons. Alternatively, it should be given uniformly, i.e. for the other species (see Frankliniella occidentalis in line 67). Please check this for the other quoted species as well.

ad 430: read: Paspalum dilatatum

Not completed citations: 550-551 (PhD Thesis?), 555-556, 575-576

Please write in Italic: 532

Typing and character errors (extra space etc.) : 31, 39, 420, 425, 511, 514, 520,

Reviewer #2: Very nice work to identify the enzymes involved in response to alfalfa weevil feeding

Point wise comments and suggestions given on attached manuscript

Regression or correlation matrix table might be much better to understand when in numbers

Numerical data would be much easy to understand if tables of the figures are presented and AVOVA for overall enzymes of respective group may be presented instead of figure for each to identify the impact of factors separately and in interaction either in manuscript or in supplementary file

The percentage comparison calculation may also be presented in method as well. Its overall correlation can also help for better relate the things or parameters in focus

6. PLOS authors have the option to publish the peer review history of their article (what does this mean?). If published, this will include your full peer review and any attached files.

Reviewer #1: No

Reviewer #2: No

---

## [Author Response · Author response to Decision Letter 0]

1 Sep 2024

Journal Requirements:

Response: We have completed the formatting revision of the manuscript according to the requirements of the PLOS ONE, which fully meets the style of the PLOS ONE.

2. Thank you for stating the following financial disclosure: " National Modern Agriculture Industry Technology System (CARS-34) of the Ministry of Agriculture and Rural Affairs".

Response: We had added the Role of Funder statement in our cover letter.

Response: I have completed the registration and updated my personal information in the system. ORCID iD is 0009-0005-7290-0600.

Response: Our supporting information is the method for measuring enzyme activity, which has been supplemented at the end of the main text.

Additional Editor Comments:

The manuscript described about the feeding activity of alfalfa leaf weevil larvae and adults, and physiological changes in alfalfa. The study design and manuscript preparation was well. Study background information, objective and methodology was clearly described in the manuscript. However, results and interpretations sections need revision. Authors needs to consider the following comments to improve the manuscript.

Response: Thank you for your positive feedback and for raising this important point. We had carefully revised the entire text and further improved the writing quality of the manuscript.

Line 29: In addition to cow, add two more examples.

Response: Thank you for the constructive feedback. We had added more examples.

Line 105: Write expansion of PBS.

Response: Thank you for your comments and suggestions on our manuscript. We had added revised it.

Line 426: Remove “[34]”

Response: Thank you for the constructive feedback. We had added removed it.

Methodology for each enzyme assay may describe in the methodology section instead of providing as supplementary section. Supportive tables/figures/details may provide as supplementary information.

Response: Thank you for your insightful comment. The enzyme activity assays were conducted using reagent kits, and the reagents in each kit for enzyme activity vary greatly. If all methods are included in the main text, it will increase the length of the text, so we have included them in the supporting information.

Add methodology for Principal Component analysis.

Response: The principal component analysis method has been added to the “Data Statistics and Processing” section.

In section 3.1. and 3.2.: Avoid repetitions of “in alfalfa leaves induced by adult alfalfa leaf weevils” and “in alfalfa leaves induced by 3rd-instar larvae of alfalfa leaf Weevils” in subsection titles.

Response: Thank you for your valuable suggestion. We had added revised it.

In Figure 2 - 5: Remove abbreviations in Y axis titles.

Response: Thank you for your comments and suggestions on our manuscript. We had removed the abbreviations in Y axis titles of Fig 2-5.

In Figure 6 – 7 & 12 - 13: Write expansions for all the abbreviations in the figure title.

Response: Thank you for your comments and suggestions on our manuscript. In Figure 6 – 7 & 12 – 13, the abbreviations in the figures have appeared multiple times in the main text, and we provided full spelling for each abbreviation when it first appeared. Therefore, we believe it is not necessary to annotate the full spelling of abbreviations in the figure title.

In Figures: Results are expressed in enzyme specific activity based units. Whereas in the text, results were described in percentage levels. Please check and describe clearly with respect to the presented results or add additional informations for better understanding.

Response: The percentage comparison is the result of comparing enzyme specific activity based units, without any special calculation.

Please check the cited references with the reference list vice versa. Check and write all the references as per the journal guidelines. For example, see line 507-508, 526-527, 546, 547, etc.,

Response: Thank you for your comments and suggestions on our manuscript. We had revised it.

Please check the typographical errors throughout the manuscript.

Response: Thank you for your comments and suggestions on our manuscript. We had revised it.

5. Review Comments to the Author

Reviewer #1: Cohesive, beautifully structured MS. The choice of topic is also an economically important field, but the stress physiology of insect damage to alfalfa is also important in basic research. The methods and implementation of the experimental design and research work are of scientific standard. The text of the thesis is easy to follow, its structure is based on a nice internal logic. Perhaps some of the groundwork ideas from Discussion could have been included in Introduction. In evaluating the results, correlations are sought and found. Unfortunately, many sentences in the conclusions chapters are identical to some sentences in the abstract and discussion chapters. Please check this for the other quote species as well.

Response: Thank you for your positive feedback and for raising this important point. We had carefully revised the entire text and further improved the writing quality of the manuscript.

Other comments:

ad 89-90: Do I need a company name? Hoagland solution is well-known, easy to prepare, widely used.

Response: Thank you for your valuable feedback. We had deleted it.

ad 160 & 284: Proposed simplification: metabolic enzymes

Response: Thank you for your insightful comment. We had revised it.

ad 190: Is this citation really relevant ?

Response: Thank you for the constructive feedback. We deleted that citation.

ad 198: TI or TPI?

Response: Thank you for your kindly suggestions. It should be “TI”, we had revised it.

AD 367/375/412/415: It is not customary to put the species descriptive auctor in parentheses. It is enough to specify the auctor at the first summons. Alternatively, it should be given uniformly, i.e. for the other species (see Frankliniella occidentalis in line 67). Please check this for the other quoted species as well.

Response: Thank you for your kindly suggestions. We had deleted the species descriptive.

ad 430: read: Paspalum dilatatum

Response: Thank you for your kindly suggestions. We had revised it.

Not completed citations: 550-551 (PhD Thesis?), 555-556, 575-576

Response: Yes, it is PhD thesis.

Please write in Italic: 532

Response: Thank you for your kindly suggestions. We had revised it.

Typing and character errors (extra space etc.) : 31, 39, 420, 425, 511, 514, 520,

Response: I am very sorry for the mistake, we had revised it.

Reviewer #2: Very nice work to identify the enzymes involved in response to alfalfa weevil feeding

Point wise comments and suggestions given on attached manuscript

Response: Thank you for your valuable feedback.

Regression or correlation matrix table might be much better to understand when in numbers

Numerical data would be much easy to understand if tables of the figures are presented and AVOVA for overall enzymes of respective group may be presented instead of figure for each to identify the impact of factors separately and in interaction either in manuscript or in supplementary file

Response: Thank you for your kindly suggestions. We had revised it in Fig 6 and Fig 12.

The percentage comparison calculation may also be presented in method as well. Its overall correlation can also help for better relate the things or parameters in focus

Response: The percentage comparison is the result of comparing enzyme specific activity based units, without any special calculation.

Line 426 Need to follow format of the journal.

Response: We have completed the formatting revision of the manuscript according to the requirements of the PLOS ONE, which fully meets the style of the PLOS ONE.

---

## [Decision Letter · Decision Letter 1]

30 Sep 2024

PONE-D-24-23169R1Alfalfa leaf weevil larvae and adults feeding induces physiological change in alfalfa enzymesPLOS ONE

Dear Dr. Liu,

Thank you for submitting your manuscript to PLOS ONE. After careful consideration, we feel that it has merit but does not fully meet PLOS ONE’s publication criteria as it currently stands. Therefore, we invite you to submit a revised version of the manuscript that addresses the points raised during the review process.

We look forward to receiving your revised manuscript.

Kind regards,

S Ezhil Vendan, Ph.D

Academic Editor

PLOS ONE

**Journal Requirements:**

**Additional Editor Comments:**

The revised manuscript and authors responses are satisfactory. However, the manuscript need minor revision with respect to the following comments;

Line 4-5: Manuscript title should be specific to the study. Authors may write “defensive enzymes of alfalfa” instead of “alfalfa enzymes” in the title.

Line 32-33: Remove the sentence “Adults feed on alfalfa stems and leaves, whereas larvae feed on leaf flesh and veins.”.

Line 33-34 & 476-478: The study conclusion is not clear. Please check “PPO is distributed in flower organs, meristems, leaves, stems, and roots ………”. Among the examined all enzymes, how PPO only played important role?. Please justify with other enzymes or write conclusion in comparative way with other enzymes.

Line 67: Is it “Phenylalaninammo nialyase” or “Phenylalanine ammonia lyase”?. Please check.

Line 73-74: Write common of Frankliniella occidentalis.

Line 88-89: Remove the sentence “This study provides a reference basis for research on alfalfa insect resistance mechanisms and breeding.” from the study objective.

Line 119-120: Write like “A. Adult feeding, B. Larvae feeding” instead of “A for adult and B for larvae”. Remove “(replicates)” from the figure legend.

Line 125-126: Add methodology for Principal Component analysis. The sentence “The perform correlation analysis using Pearson correlation coefficient.” is not clear. Please check and revise.

Reviewers' comments:

Reviewer's Responses to Questions

**Comments to the Author**

1. If the authors have adequately addressed your comments raised in a previous round of review and you feel that this manuscript is now acceptable for publication, you may indicate that here to bypass the “Comments to the Author” section, enter your conflict of interest statement in the “Confidential to Editor” section, and submit your "Accept" recommendation.

Reviewer #1: All comments have been addressed

Reviewer #2: All comments have been addressed

2. Is the manuscript technically sound, and do the data support the conclusions?

Reviewer #1: Yes

Reviewer #2: Yes

3. Has the statistical analysis been performed appropriately and rigorously? 

Reviewer #1: Yes

Reviewer #2: Yes

4. Have the authors made all data underlying the findings in their manuscript fully available?

Reviewer #1: Yes

Reviewer #2: Yes

5. Is the manuscript presented in an intelligible fashion and written in standard English?

Reviewer #1: Yes

Reviewer #2: Yes

6. Review Comments to the Author

**Reviewer #1: **Comments and suggestions from reviewers have been accepted, but not always improved (e.g. Wu 2022, Ren 2020, Malefo et al. 2020), although these changes may be subject to editorial decision.

**Reviewer #2:** (No Response)

7. PLOS authors have the option to publish the peer review history of their article (what does this mean?). If published, this will include your full peer review and any attached files.

Reviewer #1: No

Reviewer #2: No

---

## [Author Response · Author response to Decision Letter 1]

5 Oct 2024

Reviewers' comments:

Reviewer #1: Comments and suggestions from reviewers have been accepted, but not always improved (e.g. Wu 2022, Ren 2020, Malefo et al. 2020), although these changes may be subject to editorial decision.

Response: Thank you for the constructive feedback. In our manuscript, we cited two Chinese doctoral dissertations. We have traced their published English articles and replaced.

---

## [Editor Report · Decision Letter 2]

10 Oct 2024

Alfalfa leaf weevil larvae and adults feeding induces physiological change in defensive enzymes of alfalfa

PONE-D-24-23169R2

Dear Dr. Hui Liu,

We’re pleased to inform you that your manuscript has been judged scientifically suitable for publication and will be formally accepted for publication once it meets all outstanding technical requirements.

Kind regards,

Dr. S. Ezil Vendan, Ph.D

Academic Editor

PLOS ONE

---

## [Editor Report · Acceptance letter]

31 Oct 2024

PONE-D-24-23169R2 

PLOS ONE

Dear Dr. Liu, 

I'm pleased to inform you that your manuscript has been deemed suitable for publication in PLOS ONE. Congratulations! Your manuscript is now being handed over to our production team.

Kind regards, 

on behalf of

Dr. S Ezhil Vendan 

Academic Editor

PLOS ONE